# Investigation on the Formation of Cr-Rich Precipitates at the Interphase Boundary in Type 430 Stainless Steel Based on Austenite–Ferrite Transformation Kinetics

**Tao Jia * , Run Ni, Hanle Wang, Jicheng Shen and Zhaodong Wang**

The State Key Lab of Rolling and Automation, Northeastern University, Shenyang 110819, China;
RunNi3017@163.com (R.N.); stuwanghanle@aliyun.com (H.W.); shenjicheng@163.com (J.S.);
zhdwang@mail.neu.edu.cn (Z.W.)
*   Correspondence: jiatao@ral.neu.edu.cn or tao.jia.81@gmail.com; Tel.: +86-24-8368-1190

**Abstract:** The Cr-rich precipitates at the interphase boundary in stainless steels not only lead to the sensitization, which further induces the intergranular corrosion and intergranular stress corrosion cracking, but also significantly deteriorate the ductility and toughness. In this work, the formation of Cr-rich precipitates at the interphase boundary in type 430 stainless steel was investigated from the perspective of austenite–ferrite transformation kinetics. Cyclic heat treatment was firstly conducted to reveal the kinetic mode of transformation behavior, i.e., local equilibrium or para equilibrium. Subsequently, interrupted quenching during continuous cooling was carried out, which illustrated clearly the relevance of the formation of interphase Cr-rich precipitates to the Cr enrichment adjacent to the interphase boundary as revealed by line scanning of energy dispersive spectroscopy (EDS). Finally, this enrichment of Cr was interpreted by DICTRA simulation, which is based on the determined kinetic mode for austenite–ferrite transformation. This work has, for the first time, established the correlation between the formation of interphase Cr-rich precipitates and the austenite–ferrite transformation kinetics.

**Keywords:** transformation kinetics; local equilibrium; para equilibrium; Cr-rich precipitate; interphase boundary; type 430 stainless steel

---

## 1. Introduction

Intergranular corrosion (IGC) and intergranular stress corrosion cracking (IGSCC) are the main corrosion modes of stainless steels when exposed to an aggressive environment. They have been long recognized to be induced by the boundary sensitization, i.e., the existence of a chromium (Cr) depleted zone adjacent to boundaries [1–3]. Even though other underlying mechanisms [4–6] for the formation of a Cr-depleted zone are found, the precipitation of Cr-rich carbide and nitride at boundaries is certainly the major one [7,8]. Provided a sufficient chemical driving force for precipitation, this is conceivable since the interface energy for nucleation is comparatively large at boundaries and the subsequent growth would drain Cr atoms from neighboring areas alongside the boundaries [9]. Thus, from the kinetic perspective, the precipitation of Cr-rich precipitates would be easier in ferritic stainless steel (FSS) [10] or at the ferrite side of the interphase boundary in duplex stainless steel (DSS) [11] due to the lower solubility of C and N and the fast diffusivity of Cr in the ferrite phase.

In contrast to the well-investigated IGC and IGSCC, the loss in ductility and toughness caused by the Cr-rich precipitates at boundaries has drawn much less attention. Shankar et al. [12] attributed the deterioration of ductility in 316LN stainless steel to the Cr-rich precipitates at grain boundaries, their interaction with dislocations, and the associated stress buildup at the grain boundaries. Ghosh [13]

found that the fracture mode changed from transgranular to intergranular with increasing formation of grain boundary precipitates, and the ductility and fracture toughness decrease significantly. Hilders et al. [14] also related the decrease in toughness of 304L stainless steel to the increasing volume fraction of voids nucleated at the grain boundary precipitates formed during sensitization. Kumar Subodh and Shahi [15] revealed the detachment at heavily precipitated grain boundaries in the heat-affected zone of AISI 304L welds after post-weld thermal aging.

Considering the adverse effect on the in-use properties of stainless steels, investigation on the mechanism for the formation of Cr-rich precipitates would be of great importance. The present work focuses on two aspects, i.e., the austenite–ferrite transformation kinetics and the formation of Cr-rich precipitates at prior austenite/ferrite interphase boundaries in type 430 stainless steel. The experimental studies and DICTRA simulation have, for the first time, enabled the establishment of the correlation between these two physical metallurgical behaviors.

## 2. Materials and Experiments

Two steels obtained from a steel company, i.e., type 430 (8 mm hot-rolled plate) and 410S (6 mm hot-rolled plate) stainless steel were used in this study. Type 410S stainless steel was selected as a comparison for the investigation of continuous cooling transformation kinetics. Their chemical compositions are listed in Table 1.

**Table 1.** Chemical composition of experimental steels (wt %).

| Stainless Steel | C | N | Si | Mn | Cr | Ni |
|---|---|---|---|---|---|---|
| 430 | 0.04 | 0.04 | 0.25 | 0.32 | 16.32 | 0.16 |
| 410S | 0.026 | 0.025 | 0.28 | 0.27 | 12.6 | 0.15 |

Similar constituent phases appear on both phase diagrams in Figure 1 which include ferrite, austenite, chromium nitride, and carbide with body-centered cubic (BCC), face-centered cubic (FCC), hexagonal close packed (HCP), and $M_{23}C_6$ crystal structure, respectively. The enlarged lower left region of the phase diagram is shown in the inset. Chromium nitride and carbide would precipitate at ~850 °C and below. Formation of cementite is thermodynamically unfavorable. The most noticeable difference in the phase diagram was the austenite single phase region between 901 and 1037 °C in type 410S stainless steel while the maximum volume fraction of austenite in the dual phase region is 44.7% in type 430 stainless steel.

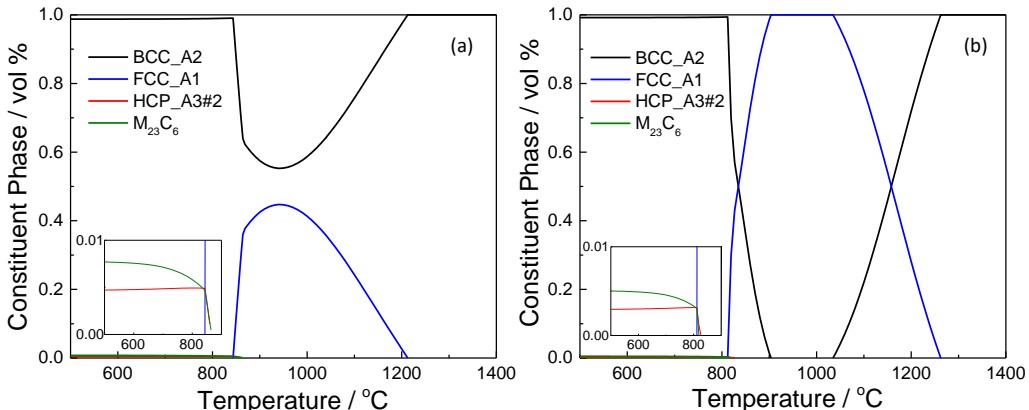

**Figure 1.** Phase diagram of type 430 (**a**) and 410S (**b**) stainless steels.

The heat treatment experiments in this work were conducted on a DIL 805A/D dilatometer (TA Instruments, New Castle, DE, USA). The sample size was Φ 4 mm × 10 mm. After machining, samples were all homogenized in a sealed quartz tube at 1200 °C for 120 min. Figure 2 shows the employed

heat treatment procedure. The sample was firstly held at 1200 °C for 5 min. Then, in the cyclic heat treatment where austenite–ferrite transformation kinetics were studied, a 30 min isothermal holding at 950 °C was carried out to create a "ferrite + austenite" dual-phase microstructure. Subsequently, one cycle of heating and cooling between 950 and 1150 °C was applied to the sample before quenching to room temperature. The corresponding rate of temperature change (RTC) was 10, 100, and 200 °C/min. In the continuous cooling experiment, which was targeted for the investigation on the formation of interphase Cr-rich precipitates, the sample was cooled at 30 °C/min from 1200 °C. Interrupted quenching was respectively conducted at 850 and 200 °C to examine the resulted microstructure.

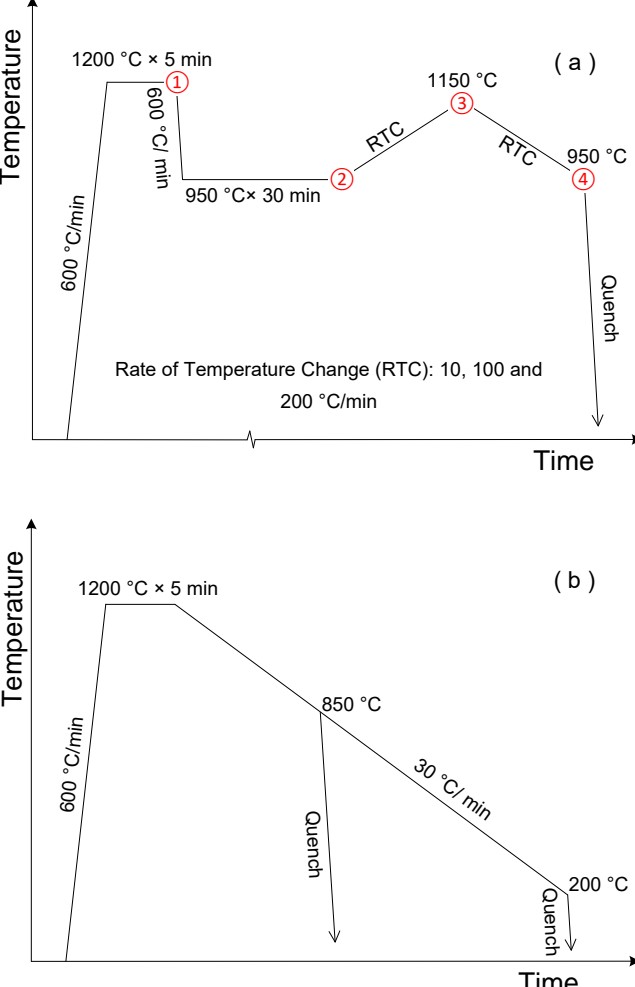

**Figure 2.** Schematic illustration of heat treatment procedure for (**a**) cyclic and (**b**) continuous cooling experiments.

Microstructure examination was made by optical microscopy (OM) and scanning electron microscopy (SEM, GeminiSEM 300, ZEISS, Oberkochen, Germany) with energy dispersive spectroscopy (EDS, Ultim Max, Oxford Instruments, Abingdon, UK). The heat-treated samples have gone through the standard metallographic preparation procedure, including grinding, polishing, and etching with 2% Nital solution.

## 3. Modeling of Austenite–Ferrite Transformation Kinetics

Over the past few decades, extensive research has been devoted to the study of austenite–ferrite transformation kinetics. Among the several proposed theories, diffusion-controlled theory is the most

important one. For the Fe-C-M (M stands for the substitutional element) system, there are two proposed thermodynamic equilibrium conditions, i.e., para equilibrium (PE) and local equilibrium (LE).

### 3.1. Para Equilibrium

PE [16] describes the equilibrium state where only interstitial atoms are free to redistribute while substitutional atoms remain configurationally frozen during transformation, i.e.,

$$\frac{u_M^\alpha}{u_{Fe}^\alpha} = \frac{u_M^\gamma}{u_{Fe}^\gamma} = \frac{u_M^0}{u_{Fe}^0} \tag{1}$$

where $u_{Fe}$ and $u_M$ are molar fractions of Fe and M with respect to substitutional sites which are termed u-fraction. The superscript 0 stands for bulk concentration and $\alpha$ or $\gamma$ denotes the ferrite or austenite phase. PE is a constrained equilibrium which is defined as

$$\begin{cases} \mu_C^\alpha = \mu_C^\gamma \\ \left(\mu_{Fe}^\gamma - \mu_{Fe}^\alpha\right) + \frac{u_M^0}{u_{Fe}^0}\left(\mu_M^\gamma - \mu_M^\alpha\right) = 0 \end{cases} . \tag{2}$$

### 3.2. Local Equilibrium

In LE [17], the chemical potential $\mu$ of carbon and substitutional element across the interface is constant, i.e.,

$$\mu_i^\alpha = \mu_i^\gamma \tag{3}$$

where $\mu$ is the chemical potential; the subscript i represents C or M. Due to the large difference in diffusivity between C and M, the LE is further classified into two types: negligible partition local equilibrium (NPLE) and partition local equilibrium (PLE). The Fe-C-Cr system, where Cr is a ferrite stabilizer, is used here for further elaboration.

In NPLE, as shown in Figure 3, the specific tie-line always connects the product phase with the Cr concentration of uCr0. Therefore, the product phase achieves the same Cr content as that in the bulk of the parent phase, and a positive or negative "spike" exists in front of the moving interface. When the interface is under NPLE, only local redistribution of Cr is required, and the transformation kinetics are controlled by carbon diffusion in the parent phase.

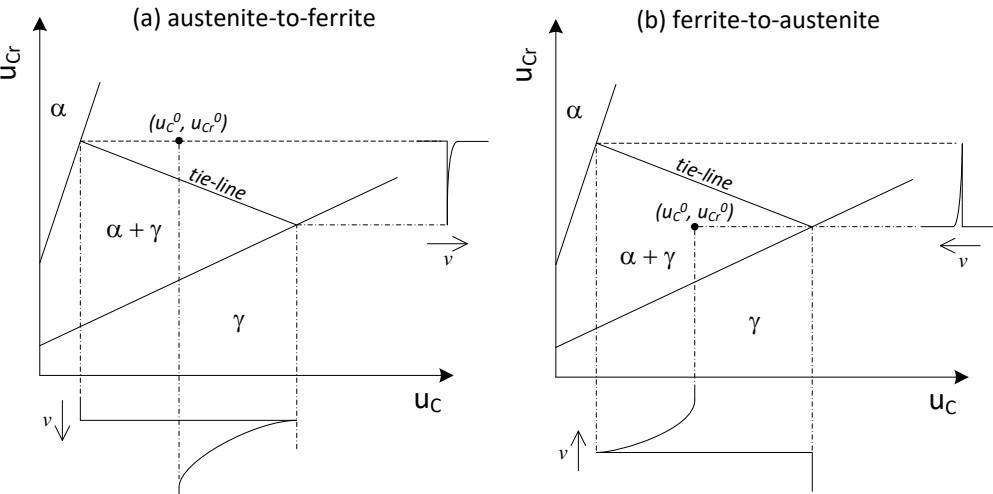

**Figure 3.** Schematic isotherm and concentration distributions depicting the (**a**) austenite-to-ferrite; (**b**) ferrite-to-austenite transformation under the negligible partition local equilibrium (NPLE) condition.

On the contrary, when partition of Cr takes place between the parent and product phase, as shown in Figure 4, long diffusion of Cr in the parent phase is necessary while a constant carbon activity is achieved from the interface to the bulk of the parent phase. In this case, the transformation is under PLE mode and the sluggish diffusion of Cr in the parent phase becomes the decisive step in controlling the kinetics.

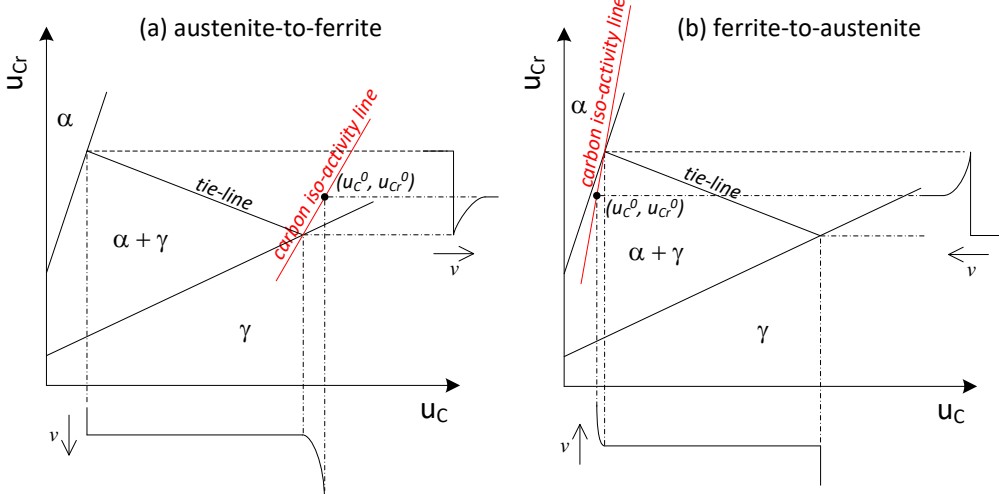

**Figure 4.** Schematic isotherm and concentration distributions depicting the (**a**) austenite-to-ferrite; (**b**) ferrite-to-austenite transformation under the partition local equilibrium (PLE) condition.

## 4. Results and Discussion

### 4.1. Determination of Austenite–Ferrite Transformation Kinetic Mode

Figure 5 shows the interruptedly quenched microstructure of type 430 stainless steel from the cyclic heat treatment with the RTC of 10 °C/min. The observed martensite, as indicated by the arrow, was transformed from the prior austenite by quenching. It is seen that, at each stage, i.e., ①, ②, ③, and ④ in Figure 2a, the prior austenite existed in the form of strips or islands in the ferritic matrix. The measured area fraction of austenite phase at ①, ②, ③, and ④ is 13.6%, 27.3%, 12.8%, and 23.7%, respectively. When the RTC increased to either 100 °C/min or 200 °C/min, the area fraction of austenite at the end of cyclic heat treatment decreased largely to about 16.5%, as shown in Figure 6, which suggests the characteristic RTC dependence of phase transformation kinetics.

To interpret the observed results, DICTRA simulation [18–20] under the assumption of one-dimensional planar geometry was carried out, where Tcfe9 thermodynamic and Mob4 mobility databases were used. In order to reduce the amount of calculation while ensuring the quality of simulation, type 430 stainless steel was simplified to the Fe-C-N-Cr system. The simulation was initiated from the beginning of isothermal holding at 950 °C and a domain size of 50 μm was used. The equilibrium constituent phases at 1200 °C were set as the starting point, i.e., ferrite and austenite with a chemical composition of Fe-0.039C-0.038N-16.34Cr (wt %) and Fe-0.118C-0.198N-14.837Cr (wt %), respectively. The initial ferrite/austenite interface was located globally at 49.35 μm. The simulation was carried out under both LE and PE conditions.

Figure 7 presents the evolution of a Cr profile during the heating and cooling stage under the LE condition. By the end of isothermal holding at 950 °C, the Cr profile exhibited a zigzag shape at the interface position, suggesting partitioning behavior of Cr from austenite to ferrite. During heating to 1150 °C, the zigzag shape of the Cr profile shrank when the interface migrated towards the austenite region. Even though the rate of change in the Cr gradient at the interface decreases with the increasing of the heating rate, a negative Cr spike in front of the moving interface was formed by the end of the heating stage, i.e., at 1150 °C, indicating a shift in transformation kinetics from a slow PLE mode to a

fast NPLE mode. A similar ferrite/austenite interface location was achieved irrespective of the heating rate. In contrast, the enrichment of Cr at the ferrite side and the depletion of Cr at the austenite side of the interface gradually built up when the interface was moving backward during the cooling stage, suggesting the transformation kinetics switched from fast NPLE mode to slow PLE mode. At this stage, the cooling rate exerts a noticeable effect on the interface migration since the diffusion of Cr is very time-consuming compared with that of C. Finally, the one-dimensional austenite fraction, which is defined as the length of the austenite region divided by total domain size, reached 29.1%, 24.3%, and 23.6% at the RTC of 10, 100, and 200 °C/min, respectively.

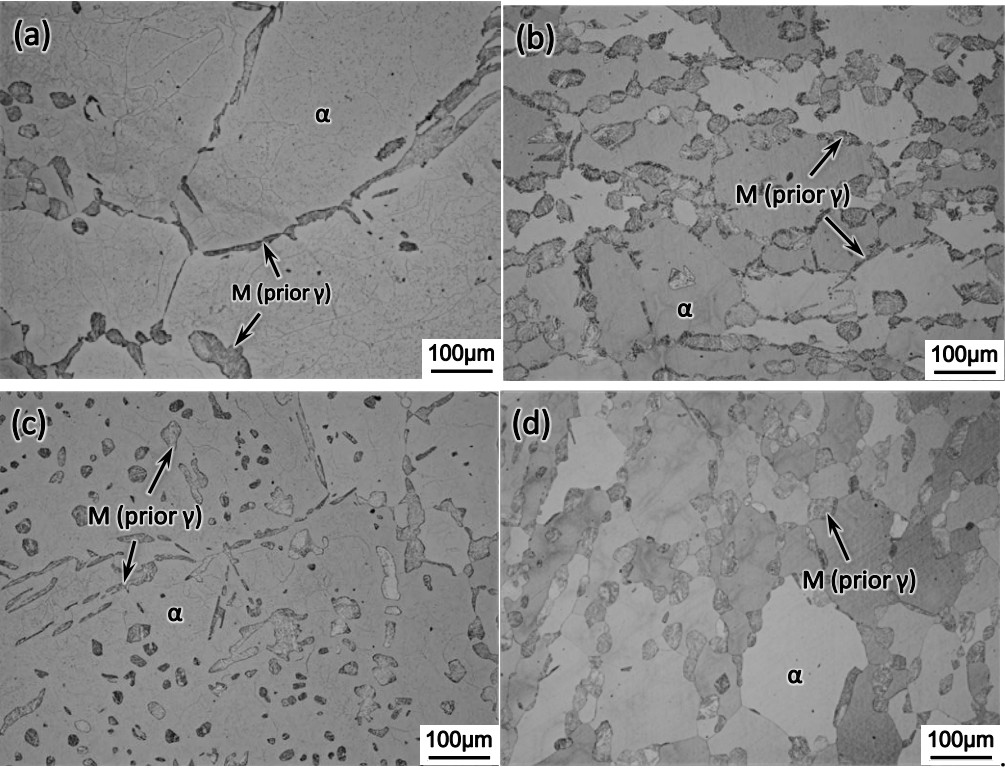

**Figure 5.** Optical micrograph showing the microstructure of an interruptedly quenched sample at (**a**) ①; (**b**) ②; (**c**) ③; (**d**) ④ during the cyclic heat treatment with the RTC of 10 °C/min.

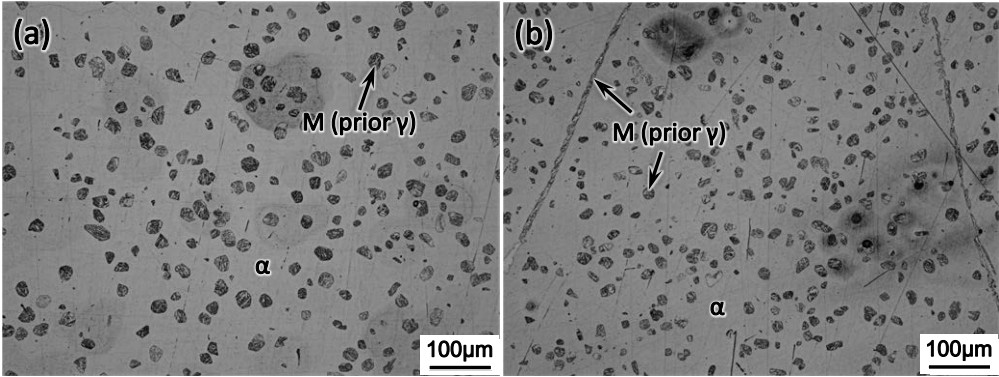

**Figure 6.** Optical micrograph showing the final microstructure of cyclic heat treatment with the RTC of (**a**) 100 °C/min; (**b**) 200 °C/min.

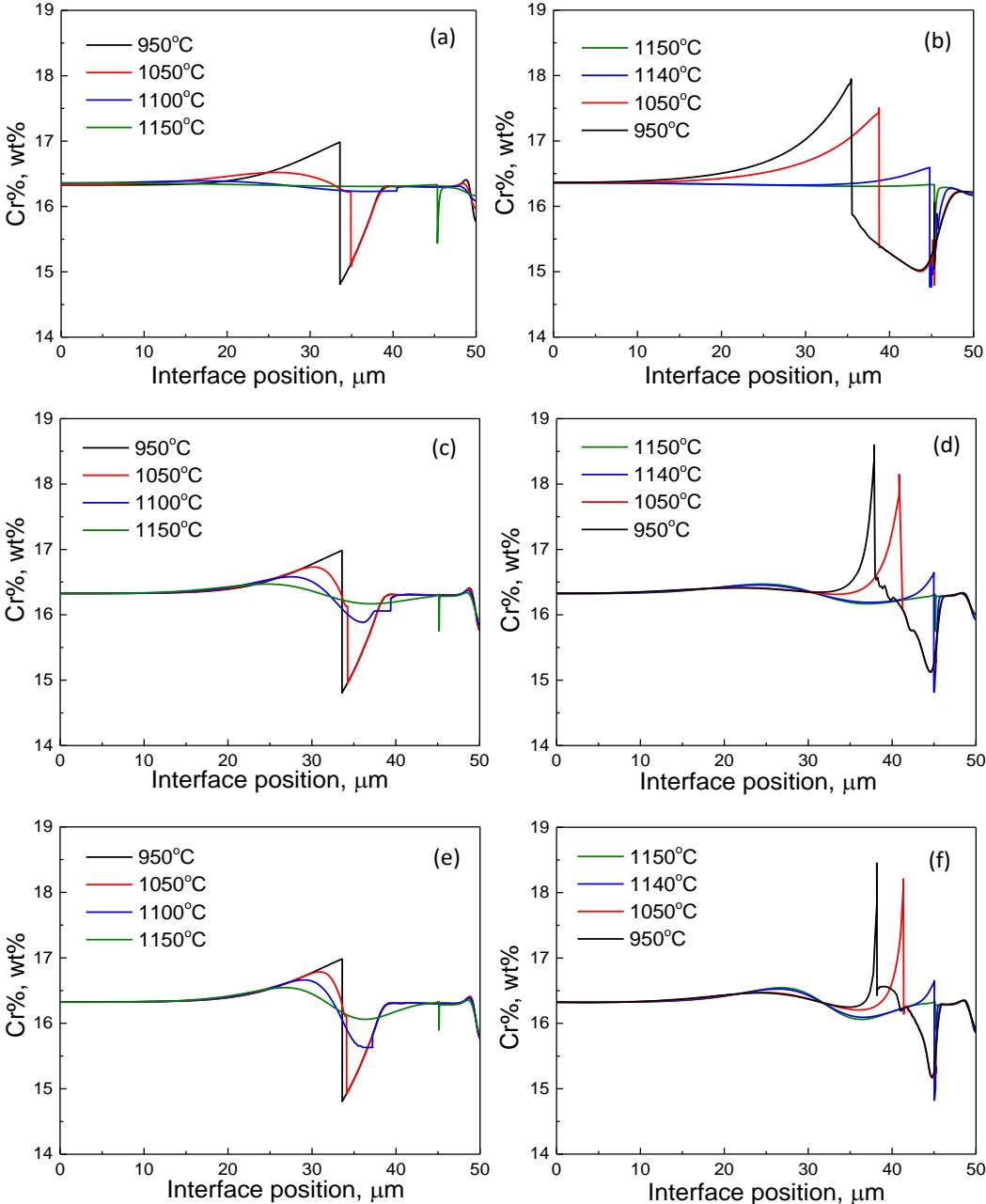

**Figure 7.** The evolution of the Cr profile during cyclic heat treatment at the RTC of (**a**,**b**) 10 °C/min; (**c**,**d**) 100 °C/min, and (**e**,**f**) 200 °C/min, where (**a**,**c**,**e**) and (**b**,**d**,**f**) correspond to the heating and cooling stage, respectively.

Simulation results from the PE condition are presented in Table 2. Under the PE condition, carbon diffusion plays a determining role for interface migration while the substitutional element Cr does not redistribute among ferrite and austenite at the interface. Therefore, the RTCs employed in this study have negligible effect on the transformation kinetics. Results from experiments and DICTRA simulation are all summarized in Table 2. It is seen that, when the RTC increases from 10 to 200 °C/min, the one-dimensional austenite fraction at the end of the cyclic heat treatment from PE simulation decreases marginally by 0.4%, in contrast to the noticeable decrease of 7% from LE simulation. From the above experimental study and DICTRA simulation, one could summarize that the transformation kinetics in type 430 stainless steel can be better captured by the simulation under the LE condition even though the Fe-C-N-Cr system is only a simplified representative of type 430 stainless steel.

**Table 2.** Measured and simulated austenite fraction by the end of cyclic heat treatment in type 430 stainless steel.

| Stainless Steels | Value Type | 10 °C/min | 100 °C/min | 200 °C/min |
|---|---|---|---|---|
| 430 | Measured [1] | 23.7% | 16.3% | 16.7% |
| Fe-C-N-Cr system | simulated (LE) [2] | 29.1% | 24.3% | 23.6% |
| | simulated (PE) [2] | 28.4% | 28.2% | 28.0% |

[1] area fraction; [2] one-dimensional fraction. LE: local equilibrium; PE: para equilibrium.

### 4.2. Mechanism for the Formation of Cr-Rich Precipitates at the Interphase Boundary in Type 430 Stainless Steel

Figure 8 shows the interruptedly quenched microstructure of the sample from the continuous cooling experiment, as shown in Figure 2b. Type 410S stainless steel is included here for comparison. The cooling rate employed, i.e., 30 °C/min was the same as the on-site measured value during the hot-rolling process. As the same as Figure 5, the observed martensite was transformed from the prior austenite by quenching. It is seen that, when samples were quenched at 850 °C, as shown in Figure 8a,c, ferrite and martensite were the only two constituent phases. After further slow cooling to 200 °C, the interphase precipitates as indicated by the arrow in Figure 8b were formed in type 430 stainless steel in contrast to its absence in type 410 stainless steel, as shown in Figure 8d, under the same heat treatment condition. Based on the calculated phase diagram in Figure 1, it is proposed that the Cr-rich precipitates at the interphase boundary were formed during the slow cooling process from 850 to 200 °C.

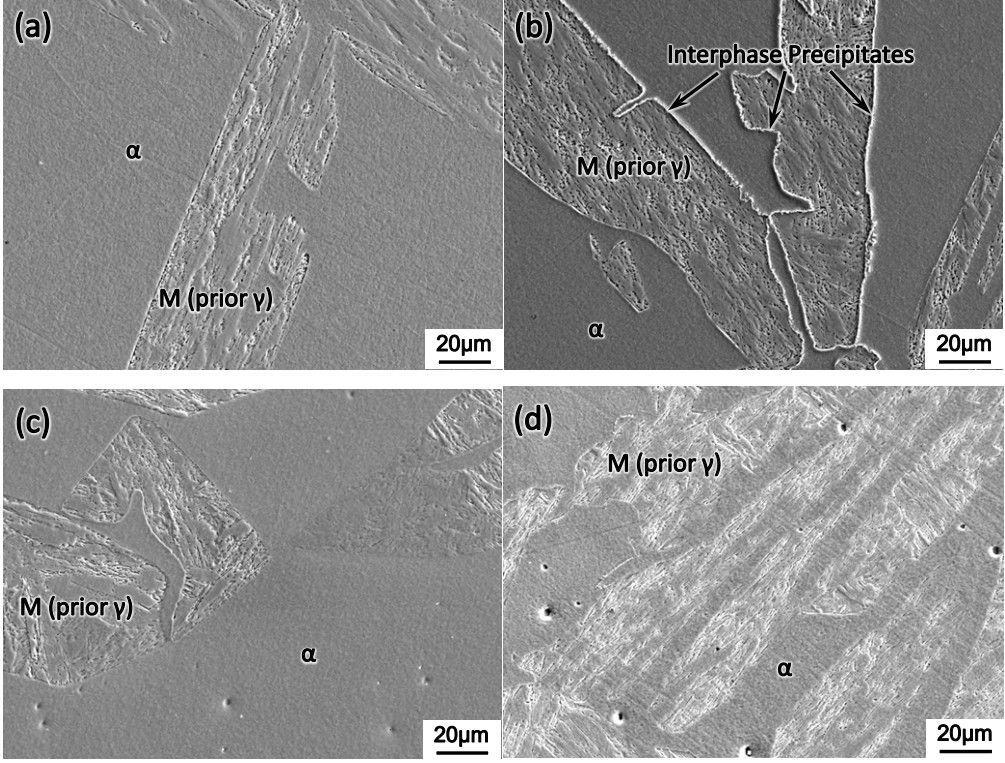

**Figure 8.** Optical micrographs showing the microstructure quenched from (**a,c**) 850 °C and (**b,d**) 200 °C, where (**a,b**) and (**c,d**) are from type 430 and 410S stainless steel, respectively.

The samples quenched at 850 °C from the continuous cooling experiment were subsequently re-examined by SEM with EDS to reveal the Cr profile across interphase boundaries. Figures 9 and 10 present the line scanning results at interphase boundaries in type 430 and 410S stainless

steel, respectively. The line scanning was conducted at a sampling rate of 6 nm/point and under a magnification of ×20,000. The black rectangular data points in Figures 9 and 10 were the raw data from line scanning. Using the "adjacent-averaging method", where 50 neighboring data points included in the adjacent 0.3 μm length line were averaged to substitute the original data point, the Cr profiles were smoothed and more clearly presented in red lines. In type 430 stainless steel, as shown in Figure 9a,b, a substantial enrichment of Cr existed in the ferrite adjacent to the interphase boundary, i.e., 17.59% relative to 15.78% at the far-end of the ferrite matrix. While, in type 410S stainless steel, as shown in Figure 10a,b, the maximum Cr% in ferrite adjacent to the interphase boundary and at the far-end of the ferrite matrix was 12.77% and 12.52%, respectively. When ferrite is enclosed by austenite, soft impingement occurs. As illustrated in Figure 9c,d and Figure 10c,d, the average Cr% in ferrite enriched to 17.2% and 12.85% in type 430 and 410S stainless steel, respectively. Thus, the formation of Cr-rich precipitates at the interphase boundaries were facilitated by the segregated Cr in type 430 stainless steel. In type 410S stainless steel, the enrichment level, if represented by the difference of Cr% in the neighboring area of interphase boundaries from the far end of the ferrite region, was much lower, i.e., 0.25% in contrast to 1.8% in type 430 stainless steel.

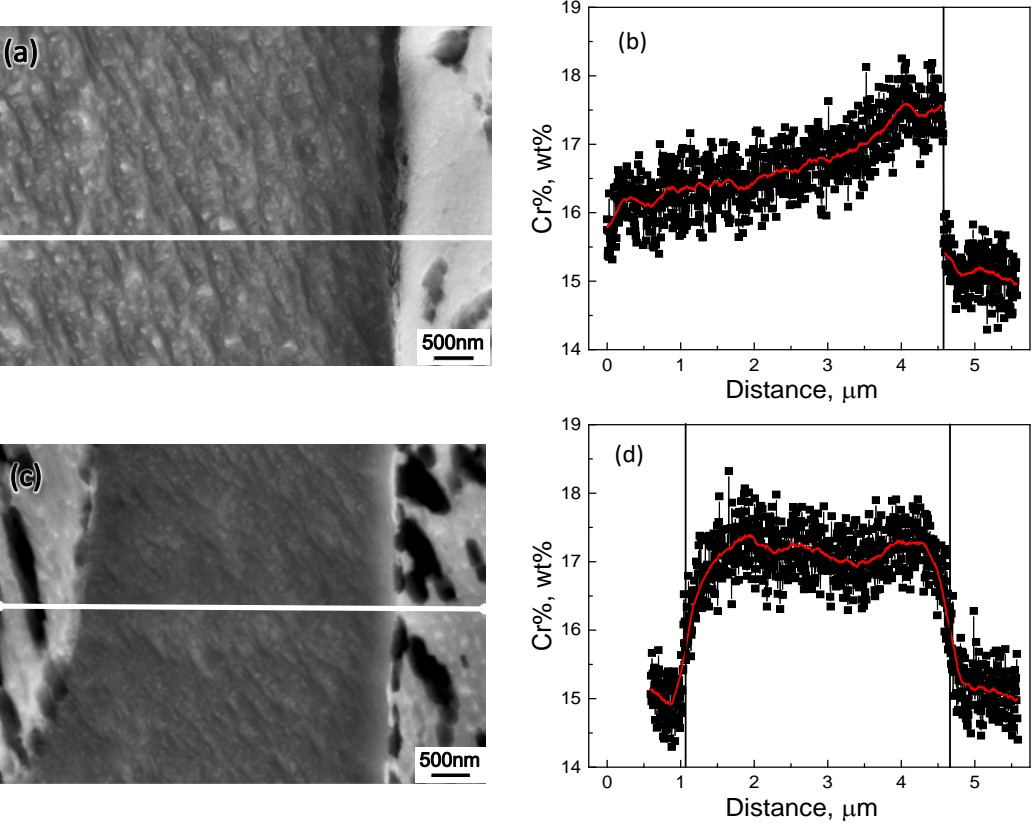

**Figure 9.** The Cr concentration profile across (**a**,**b**) single and (**c**,**d**) dual interphase boundaries in type 430 stainless steel interruptedly quenched at 850 °C.

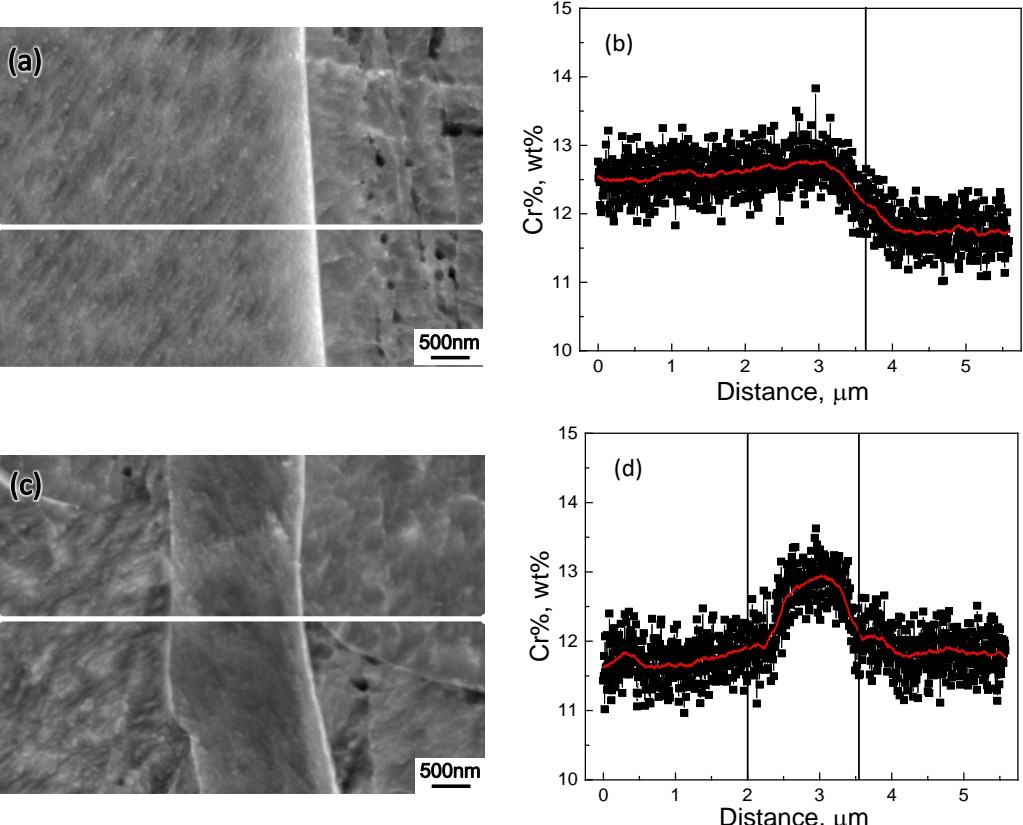

**Figure 10.** The Cr concentration profile across (**a**,**b**) single and (**c**,**d**) dual interphase boundaries in type 410S stainless steel interruptedly quenched at 850 °C.

In order to further interpret the formation of Cr enrichment, DICTRA simulation under the pre-determined LE condition in Section 4.1 is carried out where the Fe-C-N-Cr system was used as a representative of type 430 or 410S stainless steel as well. As shown in Figure 11, when the temperature decreases from 1200 to 900 °C, the interface is migrating toward the ferrite region and partitioning of Cr from austenite to ferrite can be seen. Further temperature decreases led to the backward migration of the interface and a switch of transformation kinetics to NPLE mode where a Cr spike exists in front of the interface. There are two interesting characteristics in this simulation. Firstly, the interface velocity during earlier austenite formation or the later austenite-to-ferrite transformation is much faster in type 410S stainless steel, possibly due to a large driving force as suggested by the phase diagram in Figure 1. Secondly, by the end of the simulation, a substantial Cr enrichment remains at the ferrite side of the interphase boundary in type 430 stainless steel. Compared with the line scanning results in Figures 9 and 10, an astonishing agreement has been achieved in terms of not only the shape of the Cr profile but also the Cr% in the adjacent region of the interphase boundary. Therefore, the experiment and simulation results have strongly supported the correlation between the formation of Cr-rich precipitates at the prior austenite/ferrite interphase boundary and the austenite–ferrite transformation kinetics.

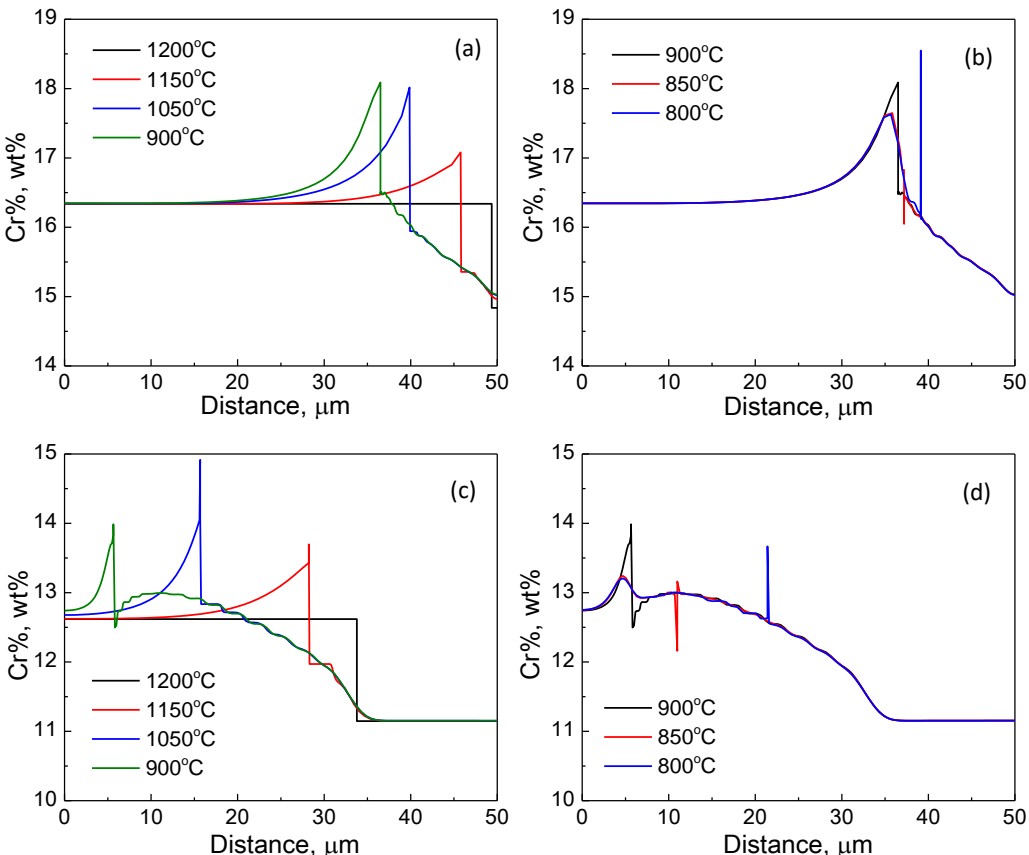

**Figure 11.** The evolution of the Cr profile during continuous cooling for type (**a,b**) 430; (**c,d**) 410S stainless steel.

## 5. Conclusions

The formation of Cr-rich precipitates at the interphase boundary in type 430 stainless steel, which not only induces intergranular corrosion and intergranular stress corrosion cracking but also significantly deteriorates the ductility and toughness, was investigated from the perspective of austenite–ferrite transformation kinetics. The following conclusions were drawn from this work.

1) The microstructure from cyclic transformation was largely affected by the rate of temperature change, which is in well accordance with the DICTRA simulation of austenite–ferrite transformation under the LE condition.

2) In contrast to type 410S stainless steel, a noticeable enrichment of Cr adjacent to the interphase boundary which facilitated the formation of interphase Cr-rich precipitates in type 430 stainless steel was revealed by EDS analysis and interpreted by DICTRA simulation under the LE condition. This has provided solid evidence for the correlation between the formation of interphase Cr-rich precipitates and austenite–ferrite transformation kinetics.

**Author Contributions:** Conceptualization, T.J. and J.S.; methodology, R.N. and T.J.; data curation, investigation and formal analysis, R.N. and T.J.; validation, H.W.; supervision, T.J.; writing—original draft preparation, review and editing, T.J.; resources, J.S. and Z.W.

**Funding:** This research was funded by the National Key R&D Program of China (2017YFB0304201) and the Fundamental Research Funds for the Central Universities (N180702012).

**Conflicts of Interest:** The authors declare no conflict of interest.

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
