# Peer review of "Investigation on the Formation of Cr-Rich Precipitates at the Interphase Boundary in Type 430 Stainless Steel Based on Austenite–Ferrite Transformation Kinetics"

_metals, doi:10.3390/met9101045_

Round 1

Reviewer 1 Report

In this paper the authors present the results relating to the formation of Cr-rich precipitates at interphase boundary in type 430 stainless steel. Cyclic heat treatment was firstly conducted to reveal the kinetic mode of transformation behaviour with the formation of Cr-rich precipitates adjacent to the interphase boundary as revealed by line scanning of EDS. The thermodynamic transformations were modelled by Dictra simulation. The paper is interesting for this journal. In order to improve the quality of the paper I suggest to consider also the corrosion phenomena that occur due to the heat treatments. Since heat treatments on stainless steel are usual for different industrial applications (such as in boilers, superheaters, heat exchangers, solid oxide fuel cells interconnects and exhaust systems) the problem relating to the corrosion due to this heat treatments should be discussed. It is also important to mention the importance of the passive films alterations (Cr-depletion in the passive films that changes its corrosion properties) due to the heat treatments that induce the corrosion. According to this, I suggest to the authors to consider the following paper:

- E. Huttunen-Saarivirta, V.-T. Kuokkala, and P. Pohjanne, Corrosion Science, 87, 344 (2014).

- F. Di Franco, A. Seyeux, S. Zanna, V.Maurice, and P. Marcus, “Effect of High Temperature Oxidation Process on Corrosion Resistance of Bright Annealed Ferritic Stainless Steel,” J. Electrochem. Soc., 164, C869 (2017).

- Did authors investigate by Polarization Curves, Electrochemical Impedance Spectroscopy the corrosion behaviour of the samples after these heat treatments?

Reviewer 2 Report

Review of “Investigation on the formation of Cr-rich precipitates at interphase boundary in type 430 stainless steel based on austenite-ferrite transformation kinetics”

The above paper present details of the formation of Cr-rich precipitates at interphase boundary in type 430 stainless steel from the perspective of austenite-ferrite transformation kinetics. The paper is quite interesting, and I believe the readers will enjoy the results.

Some comments for the authors:

For which application will be used this knowledge that are derived from this study?

The state of art is very brief and should extended

Which is the main output from this work, because I understand that the author focuses on two aspects; but what they bring new and useful for the research community?

There was mentioned chemical composition but nothing about mechanical properties. Please add them    

Figure 1 is not very clear in terms of legend (in legend are 4 elements but in Figure only 2) please rectify it

In which consideration was selected the heat treatment applied in Figure 2?

Discussion part should be validated by more literature …

From where comes “In contrast to type 410S stainless steel,” because in introduction abstract or method was not mentioned “410S stainless steel”. And why was selected to be confronted?
